# Magnitude and predictors of khat use among patients with tuberculosis in Southwest Ethiopia: A longitudinal study

**Matiwos Soboka**[1,2]*, **Omega Tolessa**[3], **Markos Tesfaye**[2,4], **Kristina Adorjan**[2,5,6], **Wolfgang Krahl**[2,7], **Elias Tesfaye**[1], **Yimenu Yitayih**[1], **Ralf Strobl**[8], **Eva Grill**[2,8]

**1** Department of Psychiatry, Jimma University, Jimma, Ethiopia, **2** Center for International Health, Ludwig Maximilian University, Munich, Germany, **3** Department of Nursing, Wolkite University, Wolkite, Ethiopia, **4** Department of Psychiatry, St. Paul's Hospital Millennium Medical College, Addis Ababa, Ethiopia, **5** Department of Psychiatry and Psychotherapy, LMU, Munich, Germany, **6** Institute of Psychiatric Phenomics and Genomics (IPPG), University Hospital, LMU Munich, Germany, **7** Department of Forensic Psychiatry, IsarAmper Klinikum, Munich, Germany, **8** Institute for Medical Information Processing, Biometrics and Epidemiology, LMU Munich, Munich, Germany

* matiwos2004@yahoo.com

## Abstract

### Introduction

Tuberculosis (TB) is a leading cause of morbidity and mortality in low and middle-income countries. Substance use negatively affects TB treatment outcomes. Our recent study has found that khat use predicted poorer adherence to anti-TB medications. However, there is scarce longitudinal study on predictors of khat use among outpatients with TB, and this study aimed at addressing this research gap.

### Methods

From October 2017 to October 2018, 268 outpatients with tuberculosis on DOTs were enrolled in a longitudinal study from 26 health institutions in Southwest Ethiopia. Structured questionnaires translated into local languages (Afaan Oromoo and Amharic) were used to assess khat use. Patients were followed for six months, and data were collected on three occasions during the follow-up. A generalized linear mixed model was used to identify the relation between khat use and predictors. Model fitness was checked using the Bayesian Information Criterion (BIC). Odds ratio (OR) and 95% CI were used to describe the strength of association between the outcome variable and predictors.

### Results

The overall prevalence of khat use at baseline and first follow up was 39.2% while it was 37.3% at second follow up. Of this, 77.1% and 96.2% of them believed that khat use reduces the side effects of anti-TB medications and symptoms of tuberculosis respectively. In the final model, being male (aOR = 7.0, p-value = 0.001), being government employee (aOR = 0.03, p-value≤0.001) and presence of alcohol use disorders (AUD) (aOR = 2.0, p-value≤0.001) predicted khat use among outpatients with tuberculosis.

**Data Availability Statement:** Data cannot be shared publicly because it is part of a mega project which is ongoing and contain sensitive patients' information. However, it would be available for

qualified researcher up on formal request to Institutional Review Board of Jimma University, Institute of Health (zeleke.mekonnen@gmail.com) and the principal investigator (matiwos2004@yahoo.com). Also, data will be shared for researchers who meet the criteria for access to confidential data.

**Funding:** The project was funded by Jimma University, Institute of Psychiatry Phenomics and Genomics, and a personal contribution from Dr. Michael Odenwald.

**Competing interests:** The authors have declared that no competing interests exist.

## Conclusion

A considerable proportion of patients with TB used khat throughout DOTs and wrongly perceived that it had health benefits. The finding implies that all patients diagnosed with TB should be screened for khat use, and a particular emphasis should be given to males and individuals with a history of alcohol use. Moreover, further studies are needed to assess patients' beliefs regarding the benefits of khat use so that interventions can be developed.

## Introduction

Globally, despite the availability of effective anti-TB drugs, tuberculosis (TB) remains a major public health problem and one of the top ten causes of death from a single agent [1]. According to the 2018 World Health Organization (WHO) report, 10 million people were infected by TB across the world, while an estimated 1.6 million of people died because of the disease in 2017 [1]. The burden of TB is exceptionally high in middle and low-income countries because of poverty, malnutrition, overcrowded living condition, poor ventilation, HIV, and other chronic diseases. Similarly, substance use disorders, remain the major contributing factors for TB in these countries [1–5]. Almost 90% of all patients with TB living in these countries face elevated TB-related mortality rates [1, 4]. Out of the total deaths attributed to TB in 2017, over 80% of the deaths were from Africa and Southeast Asia [1]. Ethiopia is one of the 22 countries with the highest burden of tuberculosis with an incidence rate of 164 and a mortality rate of 24 per 100,000 [1, 4]. Moreover, TB was the second most frequent cause of death in Ethiopia next to Malaria [4]. Non-adherence to the medication has been earmarked as one of the major issues contributing to excess mortality in Ethiopia [6–8]. Besides, non-adherence increases the risk of multi-resistant TB strains. While any substance use disorder among patients with TB might decrease adherence, it has been shown that excessive use of khat and alcohol may be one major reason for non-adherence to treatment regimens in Ethiopia [9–11].

Khat is an amphetamine-like natural stimulant that has legally been used for many years in East Africa and the Southern Arabian Peninsula [12–14]. Khat use belongs to stimulant use disorder [15]. Fresh leaves of khat contain more than 40 types of compounds, among these, cathinone and cathine are known stimulants [12–14, 16, 17].

Studies showed that people use khat to be alert while praying, to reduce the feeling of hunger, to enhance productivity at work, and to elevate their mood and to be physically strong [18–22]. Also, factors such as common mental disorder, being male, and other sociodemographic characteristics were found to determine khat use in the general population [21, 23]. However, using khat for a long period leads to khat use disorder which could have a potential impact on the mental and physical health of the users [24, 25]. Likewise, cathinone which is found in khat has been linked to a decreased immune response that might increase the risk of developing TB [26]. Besides, TB seems to be more frequently underdiagnosed in khat users [27–29]. Khat users with TB were found to have higher bacillary load and were more likely to develop drug resistance [30]. Also, they were more likely to be stigmatized [31], had longer treatment regimens [32], poor adherence [7, 19], poor appetite [33], and increased levels of anxiety [10]. Even though khat use is known to affect treatment outcomes and mental health of patients with tuberculosis, there is only limited longitudinal study on the magnitude and predictors of khat use among patients with TB in Ethiopia and other African countries. Knowing the predictors and magnitude of khat use would help to tailor interventions and to intensify the efforts to improve treatment outcomes of TB. Moreover, early identification of

predictors of khat use is important to take preventive measures to mitigate complications such as comorbid mental illness and MDR-TB. Therefore, our study aimed at assessing longitudinally the magnitude and predictors of khat use among patients with TB in Southwest Ethiopia.

## Methods

### Study area and design

A longitudinal study was conducted among patients with tuberculosis in Jimma Zone, Southwest Ethiopia. Jimma Zone has more than three million inhabitants; about 3% of the total population of Ethiopia. In Ethiopia, TB care is mainly provided by local decentralized health centers to increase take-up of therapy and to monitor Directly Observed Treatment (DOT). There were 112 health centers in Jimma Zone. Out of these, 91 were providing services for patients with TB at the time of data collection. Likewise, there were dedicated TB treatment services at all hospitals. Patients were recruited from four hospitals and 22 randomly selected health centers of Jimma Zone. Twenty health centers and three hospitals were situated in rural areas whereas one hospital and two health centers are found in Jimma town. The study was conducted over a year from October 2017 to October 2018.

**Study population and sampling procedure.** This study included all patients who had recently been diagnosed with TB and started DOT in the selected health centers and hospitals. Patients who had started TB treatment within less than four weeks and not planning to transfer to other health institutions were included in the study. There are two reasons for including new patients who started treatment within four weeks. The first one is to see if there will be any change over time and the second is to see whether the patients increase or decrease their Khat use. Patients whose age is less than 18 years, patients infected with multidrug-resistant TB strains, polysubstance users (using three or more substances), and patients who were hospitalized during data collection were excluded from the study. The data for this study was drawn from a cohort registered as //////476/2011 (Institute of health, research, and postgraduate director 476/2011) which aimed to assess substance use disorder, quality of life, mental health and adherence to anti-TB medication. The sample was calculated considering power = 80%, 95% confidence interval, 62.4% prevalence of non-adherence to anti-TB medications among khat users, 43.6% prevalence of non-adherence among non-khat user TB patients, and 20% of drop out. The total sample size was 268, and patients who fulfilled the inclusion criteria and consented to participate in the study during the data collection period were consecutively recruited then baseline data were collected. Patients were followed on two occasions: at the end of two and six months.

Patients who were using khat (105) and free of khat use (163) were followed for six months. Detail information regarding the study was given by trained data collectors to each patient before the written informed consent was obtained.

**Data collection procedure.** Before data collection, the questionnaires were pretested on a sample (5% of the total sample) of patients with TB outpatients who had been on treatment at one health center in Agaro (a town found in Jimma zone at a distance of about 45 kilometers from Jimma city) to check whether the questions work as intended or understood by patients. Fourteen patients from the pretest were not included in the final analysis of the data. Patients were interviewed on three occasions, namely, baseline (starting treatment), first follow up (after 2 months), and second follow up (at the end of six months). The follow up was made at the end of two and six months of treatment because it is the end of intensive as well as the continuous phase of the treatment. Also, at the end of the six-month, patients received another test for tuberculosis so that the status of the patients would be known. Recruitment of patients and data collection were carried out by health professionals who were working in the

tuberculosis clinic and specifically trained on the questionnaires, supervised by trained district focal persons. Interviews were conducted within the respective health institutions when the participants came for their TB clinic visits. All questionnaires including the pretest were translated into Afaan Oromoo and Amharic languages because the participants speak either of the two languages.

**Data collection tools.** *Outcome variable*. A questionnaire used to assess khat use was developed after reviewing different kinds of literature because there is no specific standard tool to assess khat use in any population. In this study, khat use was defined as using khat in the past 30 days before the interview [34]. The questionnaire includes ever use of khat, current khat use, frequency, amount, and patients' beliefs regarding khat use.

*Explanatory variables*. Socio-demographic characteristics. Structured questionnaires were used to assess the socio-demographic characteristics (age, sex, marital status, level of education, religion, ethnicity, annual income, household size, occupation, place of residence) of the participants. Income was categorized considering the minimum monthly wage for employees of a governmental organization in Ethiopia which is 1,214 Ethiopian birr (36.67 Euros) [35]. Then, the monthly income of each patient was multiplied by 12 months to obtain the annual income, and we used a cutoff 14,568 Ethiopian birr (439.98 Euros).

*Alcohol use disorders (AUDs)*. Alcohol use disorder identification test (AUDIT) was used to assess alcohol use disorders. The AUDIT has been evaluated over two decades and provides an accurate measure of the risk of AUDs across gender, age, and cultures. A multi-country validation of AUDIT among people attending primary health care in Norway, Australia, Kenya, Bulgaria, Mexico and the United States of America showed that at a cut-off score of eight or more, the sensitivity and specificity of AUDIT for AUDs were 0.90 and 0.80, respectively [36]. The AUDIT has been translated and adapted for studies in the Ethiopian setting [37].

*Disease-related factors*. Type of TB diagnoses (smear-positive, smear-negative, and extra-pulmonary TB) were collected from the patients' charts.

*Comorbidities*. Any comorbidity such as HIV, previous mental illness, hypertension, and diabetes mellitus were collected from patients' charts.

*Mental distress*. Self-reporting questionnaire-20 (SRQ-20) which was developed by WHO was used to assess mental distress. This questionnaire assesses depressive, anxiety, and somatic symptoms that patients have experienced in the past four weeks. SRQ-20 has been adapted and validated in the Ethiopian setting, but the cut-off point varies from study to study [38, 39]. In this study, a total score of below 7 indicates the absence of mental distress whereas values of 7 and above indicate mental distress. At a cut-off point 7/8 the sensitivity and specificity was 89.7% and 95.2% respectively [40, 41].

**Data analysis.** Participants' characteristics and study variables were described using descriptive statistics. A generalized linear mixed model was used to examine the predictors of khat use over six months. The model was built based on the theoretical importance and the adequate number of participants in each cell for each category. The missing value was excluded from the analysis. The findings have been adjusted for potential confounders. An intercept only model was used to investigate khat use over time (model 0) without adding other variables; model 1 investigated the longitudinal association of khat use and socio-demographic characteristics variables. Model 2 investigated the association between socio-demographic variables, mental distress, and the outcome variable (khat use). Model 3 was adjusted for the full set of predictors and examined covariates related to the khat use. Model fit was examined with the Bayesian Information Criterion (BIC). Lower BIC indicates a better model fit. Data were analyzed using R studio (1.2.1335). The study findings are reported in line with the Strengthening the Reporting of Observational Studies in Epidemiology (STROBE) statement.

**Ethical considerations.** Ethical clearance was obtained from Jimma University and LMU Ethical Review Boards. After the participants were given detail information about the importance of the study, written informed consent was obtained from each patient. The anonymity of the study participants was kept in all stages of data processing and write-up of the manuscript. Patients who were using khat more than once weekly were advised to contact a mental health professional.

## Results

### Socio-demographic and clinical characteristic

In this longitudinal study, a total of 268 patients (age range of 18 to 80 years, mean age 32.4, SD = 14.4, 60.1% male) were recruited. The majority of the study participants were married (58.6%) and Muslim (61.6%). Two-third (63.1%, n = 169) of all participants did not attend formal education (see Table 1). A total of 40.3% (n = 108), 32.5% (n = 87), and 27.2% (n = 73) were diagnosed as smear-positive, smear-negative and extrapulmonary TB, respectively. At baseline, 3.7% (n = 10) patients were diagnosed with HIV, and 7.1% (n = 19) with other comorbidities (see Table 1). There were 22 missing data of annual income which we excluded from the analysis.

**Prevalence of khat use.** The prevalence of khat use was 39.2% (n = 105) at baseline and after two months, and 37.3% (n = 100) at the end of six months. Of the total khat users, 24.8% (n = 26), 46.7% (n = 49), and 37.0% (n = 37) chewed khat daily at baseline, first follow up and second follow up respectively, while 55.2% (n = 58), 46.7% (n = 49) and 32.0% (n = 32) chewed it 2–3 times per week.

Those patients who were using khat were compared against patients free of khat use during the follow-up. Males were found to use khat more than women at baseline (46.6% versus 28.0%, p<0.05), at second month (46.6% versus 28.0%, p<0.001), and sixth month (41.6% versus 30.8%). The majority of khat users were merchants (72.4%, n = 21), followed by farmers (51.1%, n = 47), and being educated till the tertiary level (53.6%, n = 15) at baseline (see Table 1). The majority of khat users (77.1%, n = 81) believed that khat can reduce medication side effects or reduce symptoms of tuberculosis (96.2%, n = 101).

The prevalence of mental distress among patients with khat users was 42.1% (n = 69), 49.0% (n = 50) and 39.0% (n = 23) at baseline (T0), first (T1) and second (T2) follow up respectively. At the end of the second month, the majority (76.7%, n = 33), of khat users were found to have alcohol use disorder).

**Predictors of khat use.** The results of multivariable modeling are shown in Table 2. The prevalence of khat use did not change significantly over time (p = 0.48 in the final fully adjusted model). The strength of association was improved over time as witnessed by further improvement of model fit (BIC = 815.7). Being male and having AUDs were indicators for khat use after multivariable adjustment. Merchants still had a higher probability of khat use than government employees or day laborers. The odds of khat use among government employees was 97% lower when compared to that of merchants (aOR = 0.03, p-value = 0.001). Age, income, and mental distress were not associated with khat use.

## Discussion

To our knowledge, this study is the first longitudinal study investigating predictors of khat use among patients with tuberculosis in Africa. Alarmingly, TB-patients believed that the use of khat is beneficial during treatment. Alcohol use disorder and male gender were predictors of continued khat use among TB patients in Southwest Ethiopia.

**Table 1. Characteristics of patients with tuberculosis in Southwest Ethiopia 2017/18 (n = 268).**

| Variables | | Frequency N (%) | Khat use | | |
|---|---|---|---|---|---|
| | | | Baseline | First follow-up (end 2$^{nd}$ month) | Second follow-up (end of six months) |
| | | | N (%) | N (%) | N (%) |
| Gender | Male | 161(60.1) | 75(46.6) | 75(46.6) | 67(41.6) |
| | Female | 107(39.9) | 30(28.0) | 30(28.0) | 33(30.8) |
| Age | 18–24 | 93(34.7) | 33(35.5) | 30(32.3) | 28(30.1) |
| | 25–34 | 87(32.5) | 32(36.8) | 30(34.5) | 32(36.8) |
| | 35–44 | 36(13.4) | 18(50.0) | 20(55.6) | 15(41.7) |
| | 45–54 | 27(10.1) | 13(48.1) | 13(48.1) | 13(48.1) |
| | 55–64 | 25(9.3) | 9(36.0) | 12(48.0) | 12(48.0) |
| Annual income in Eth Birr | <14,568 | 206(76.9) | 86(41.7) | 85(41.3) | 83(40.3) |
| | ≥14,568 | 40(14.9) | 11(27.5) | 15(37.5) | 13(32.5) |
| Marital status | Single | 97(36.2) | 33(34.0) | 30(30.9) | 26(26.8) |
| | Married | 157(58.6) | 66(42.0) | 70(44.6) | 68(43.3) |
| | Divorced/widow | 14(5.2) | 6(42.9) | 5(35.7) | 6(42.9) |
| Religion | Orthodox | 82(30.6) | 21(25.6) | 19(23.2) | 20(24.4) |
| | Muslim | 165(61.6) | 83(50.3) | 84(50.9) | 78(47.3) |
| | Protestant and others | 21(7.8) | 1(4.8%) | 2(9.5) | 2(9.5) |
| Ethnicity | Amhara | 59(22.0) | 13(22.0) | 12(20.3) | 12(20.3) |
| | Oromo | 165(61.6) | 79(47.9) | 79(47.9) | 74(44.8) |
| | Tigre/Gurage | 44(16.4) | 13(29.5) | 14(31.8) | 14(31.8) |
| Occupation | Merchant | 29(10.8) | 21(72.4) | 7(58.6) | 17(58.6) |
| | Farmer | 92(34.3) | 47(51.1) | 47(51.1) | 49(53.3) |
| | Government Employee | 105(39.2) | 24(22.9) | 26(24.8) | 21(20.0) |
| | Daily laborer | 42(15.7) | 13(31.0) | 15(35.7) | 13(31.0) |
| Education | No formal education | 169(63.1) | 15(32.5) | 56(33.1) | 46(27.2) |
| | Primary/secondary | 71(26.5) | 35(49.3) | 34(47.9) | 36(50.7) |
| | Tertiary | 28(10.4) | 15(53.6) | 15(53.6) | 18(64.3) |
| Family size | Less than 5 | 181(67.5) | 70(38.7) | 37(42.5) | 66(36.5) |
| | 5 and more | 87(32.5) | 35(40.2) | 8(37.6) | 34(39.1) |
| Residence | Rural | 127(47.4) | 59(46.5) | 57(44.9) | 55(43.3) |
| | Urban | 141(52.6) | 46(32.6) | 48(34.0) | 45(31.9) |
| HIV | Positive | 10(3.7) | 6(60.0%) | 7(58.3) | 8(42.1) |
| | Negative | 258(96.3) | 99(38.4) | 98(38.3) | 92(36.9) |
| Mental distress | No | | 36(34.6) | 55(33.1) | 77(36.8) |
| | Yes | | 69(42.1) | 50(49.0) | 23(39.0) |

*There were 22 missing data of annual income.

In this study, the prevalence of khat use was slightly declined over time from 39.2% at baseline to 37.3% at the end of six months. The reason might be patients use more khat at baseline as a self-treatment for tuberculosis related symptoms and medication side effects. Because we identified that most patients believe khat can reduce symptoms of tuberculosis and medication side- effects. So, when they get improvement from the disease, they might reduce their khat use but further study is needed.

The prevalence of khat use observed in this study at baseline (39.2%), during the first (39.2%) and second follow up (37.3%) was far higher than another study conducted in South Ethiopia which found moderate and high khat use to be 14.3% and 1.7% respectively [9]. The

**Table 2. Predictors of khat use among patients with tuberculosis in Southwest Ethiopia (n = 268) in 2017/18).**

| Variables | | Intercept only (empty model) | | | Model1 (Socio-demography) | | | Model2 (Model 1 including mental distress) | | | Model3 (Full model) | | |
|---|---|---|---|---|---|---|---|---|---|---|---|---|---|
| | | OR | P-Value | 95%CI (upper, lower) | aOR | P-Value | 95%CI (upper, lower) | aOR | P-Value | 95%CI (upper, lower) | aOR | P-Value | 95%CI (upper, lower) |
| Gender | Female | Reference | | | | | | | | | | | |
| | Male | - | - | | 5.9 | **0.01** | 2.0,17.7 | 5.9 | **0.01** | 2.0,17.7 | 7.0 | **0.001** | 2.2,22.2 |
| Age | 18–24 | Reference | | | | | | | | | | | |
| | 25–34 | - | - | | 0.9 | 0.83 | 0.2,3.2 | 0.9 | 0.83 | 0.2,3.2 | 0.9 | 0.83 | 0.2.3.4 |
| | 35–44 | - | - | | 1.8 | 0.50 | 0.3,10.1 | 1.8 | 0.49 | 0.3,10.1 | 2.0 | 0.47 | 0.3,12 |
| | 45–54 | - | - | | 1.1 | 0.91 | 0.2,7.4 | 1.1 | 0.92 | 0.2,7.4 | 1.3 | 0.82 | 0.2,9.2 |
| | 55–64 | - | - | | 0.5 | 0.41 | 0.1,3.0 | 0.5 | 0.42 | 0.1,3.0 | 0.5 | 0.47 | 0.1.3.5 |
| Annual Income in Ethiopian birr | <14,568 | Reference | | | | | | | | | | | |
| | >14,568 | - | - | | 0.4 | 0.19 | 0.1,1.6 | 0.4 | 0.19 | 0.1,1.6 | 0.3 | 0.16 | 0.8,1.5 |
| Occupation | Merchant | Reference | | | | | | | | | | | |
| | Farmer | - | - | | 0.2 | 0.08 | 0.04,1.2 | 0.2 | 0.08 | 0.04,1.2 | 0.2 | 0.05 | 0.02,1.0 |
| | Government Employee | - | - | | 0.04 | **0.001** | 0.01,0.3 | 0.04 | **0.001** | 0.01,0.3 | 0.03 | **0.001** | 0.01,0.2 |
| | Day laborer | - | - | | 0.1 | 0.03 | 0.01,0.8 | 0.1 | **0.02** | 0.02,0.8 | 0.1 | **0.01** | 0.01,0.6 |
| Religion | Orthodox | Reference | | | | | | | | | | | |
| | Muslim | | - | | 3.3 | **0.04** | 1.0,10.6 | 3.3 | **0.04** | 1.0,10.6 | 2.6 | 0.3 | 0.8,8.6 |
| | Protestant | | - | | 0.03 | **0.01** | 0.01,0.5 | 0.03 | **0.01** | 0.01,0.5 | 0.01 | **0.01** | 0.01,0.3 |
| AUD | No | Reference | | | | | | | | | | | |
| | Yes | | - | | | | | | | | 2.0 | **0.001** | 6.0,38.1 |
| Mental distress | No | Reference | | | | | | | | | | | |
| | Yes | | | | - | - | - | 1.0 | 0.96 | 0.5,2.0 | 1.0 | 0.94 | 0.7,5.2 |
| Time | | 0.9 | 0.48 | 0.7,1.2 | 0.8 | 0.2 | 0.6,1.1 | 0.8 | 0.25 | 0.6,1.2 | 0.9 | 0.47 | 0.6,1.3 |
| BIC | | 819.0 | | | 815.2 | | | 821.9 | | | 815.7 | | |

difference might be due to the fact that the setting for the present study is known with a higher level of khat consumption than other regions and zones in Ethiopia except for the Diredawa town and Harari region [42]. In these regions, khat is considered as part of the culture for the lubrication of social cohesion. Arguably, it is also seen as a help to stay alert during the praying time which patients would not like to miss [18, 22, 43]. This is generally worrisome as khat use has been shown to be associated with poor treatment outcomes among patients with tuberculosis [44].

The prevalence of khat use found in the current study is lower than the finding from a prospective nest case-control study done in Yemen (46.7%) [19]. The difference might be due to the definition of current khat use which is 30 days in our study, while this was not specified in the Yemenite study. Likewise, socio-cultural differences might contribute to the discrepancy between the two studies. We found that the proportion of khat use among male participants (46.6%) was far higher than among female participants (28.0%), which is consistent with the findings of previous studies [45, 46]. This might be due to cultural restriction on females regarding substance use including khat [45, 46]. Even though khat consumption among women is substantially lower than men, they are at higher risk of mental and physical effects of substance use disorder than men which is associated with hormonal factors [45, 47]. In this study, more than 2/5th of khat users reported that they were suffering from mental distress. Previous studies found that substance use (khat, alcohol, and tobacco) is associated with

mental distress [10, 48], and specifically, khat was reported to increase emotional disturbance [49]. However, we could not show a clear association of mental distress and khat use in the adjusted models. Patients might be tempted to use khat as a self-treatment for their mental distress, but this observation needs to be supported by further investigations.

During the first follow up, a majority (76.7%) of patients who were using khat reported that they have alcohol use disorder. This could be due to the fact that patients might resume drinking alcohol after they begin to feel better within two to three weeks because bacterial load then usually starts to decline [4]. In addition, since alcohol counteracts the stimulant effect of khat such as sleep disturbance and restlessness, patients might be inclined to use both substances together [43, 50, 51]. Likewise, in this study, khat use was associated with alcohol use which is in agreement with previous studies [43, 48]. Combining alcohol and khat would affect patients' treatment outcomes and lead to physical and mental health problems. Because, both khat and alcohol were found to have a potential impact on the immunity of the user and associated with mental distress so that patients may develop severe medical and mental health complications or die earlier than non-users [26, 52, 53]. Furthermore, these two substances have an association with treatment-resistant tuberculosis [30, 32, 54, 55].

This study found that the majority (77.1%) of khat users believe that using khat can reduce anti-TB medication side effects which is consistent with a cross-sectional study conducted in the Butajira, Ethiopian [18]. This may be due to misinterpreting the euphoric mood from khat as a decrease in medication side effects. However, health professionals should create awareness regarding the effect of khat on mental and physical health so that patients may consider reducing their khat use. In this study, almost all khat users (96.2%) believe that khat use reduces TB symptoms, however, this is in clear contrast to the study indicating that patients with khat use had a higher bacterial load [30].

Moreover, we have found that merchants were using khat more than a farmers, government employees, and daily laborers which is in line with studies conducted in the general population [42, 56, 57]. This might be due to the fact that merchants use khat to be alert and energetic at the workplace [58]. Also, some of the merchants may be khat sellers and, as a result, they chew khat to attract customers but further study assessing this situation is needed.

To our knowledge, our study was unique in exploring predictors of continued khat use among patients with tuberculosis using a longitudinal study with multiple assessments and with no attrition. However, the following limitations need to be acknowledged. Because of social desirability, patients might minimize or deny their khat use and this could underestimate the magnitude of khat use. It could also affect the association of predictors with the outcome variables. Also, health professionals who were working in a TB clinic collected the data that might contribute to this bias; because patients may be inclined to not report about their khat use or minimize its amount. However, the prevalence of khat use was still higher than in other studies so our estimates might be rather conservative. There is no standardized instrument for the assessment of khat use. However, we are confident that we were able to capture khat use with sufficient precision. Moreover, patients with MDR-TB and who were attending their treatment at health posts, i.e. in more remote areas, were excluded from the study; hence the findings of the study may not be generalized to all patients with TB in Southwest Ethiopia. The findings of this study cannot be generalized to those patients who are getting treatment at the inpatient department who have limited access to psychoactive substances including khat. Moreover, since our sample size is not adequate, it might be difficult to draw a strong conclusion, but we can make an estimation based on the sample size without having a critical problem that could affect our findings. Also, we did not cover all possible predictors, and as a result, we recommend a qualitative study to explore other predictors of khat use among TB patients.

## Conclusions

In conclusion, a significant proportion of patients on anti-TB continue to use khat throughout their course of treatment. Predictors of khat use were being male and concomitant alcohol use disorder. These findings underscore the need to integrate the screening and treatment for substance use, specifically for khat, into the tuberculosis services. All patients diagnosed with TB should be screened for khat use and a particular emphasis should be given to males and individuals with a history of alcohol use. Furthermore, patients' beliefs about the beneficial effects of khat on tuberculosis outcomes need to be investigated so that these beliefs can be counteracted effectively.

## Acknowledgments

We are thankful for the study participants to compromising their time to participate in the study.

## Author Contributions

**Conceptualization:** Matiwos Soboka.

**Data curation:** Matiwos Soboka.

**Formal analysis:** Matiwos Soboka, Omega Tolessa, Markos Tesfaye, Kristina Adorjan, Wolfgang Krahl, Elias Tesfaye, Yimenu Yitayih, Ralf Strobl, Eva Grill.

**Funding acquisition:** Matiwos Soboka, Markos Tesfaye, Kristina Adorjan, Wolfgang Krahl, Elias Tesfaye, Yimenu Yitayih, Ralf Strobl, Eva Grill.

**Investigation:** Matiwos Soboka, Omega Tolessa, Markos Tesfaye, Kristina Adorjan, Wolfgang Krahl, Elias Tesfaye, Yimenu Yitayih, Ralf Strobl, Eva Grill.

**Methodology:** Matiwos Soboka, Omega Tolessa, Markos Tesfaye, Kristina Adorjan, Wolfgang Krahl, Elias Tesfaye, Yimenu Yitayih, Ralf Strobl, Eva Grill.

**Project administration:** Matiwos Soboka.

**Resources:** Matiwos Soboka, Omega Tolessa, Markos Tesfaye, Kristina Adorjan, Wolfgang Krahl, Elias Tesfaye, Yimenu Yitayih, Ralf Strobl, Eva Grill.

**Software:** Matiwos Soboka, Omega Tolessa, Markos Tesfaye, Kristina Adorjan, Wolfgang Krahl, Elias Tesfaye, Yimenu Yitayih, Ralf Strobl, Eva Grill.

**Supervision:** Matiwos Soboka, Omega Tolessa, Markos Tesfaye, Kristina Adorjan, Wolfgang Krahl, Elias Tesfaye, Yimenu Yitayih, Ralf Strobl, Eva Grill.

**Validation:** Matiwos Soboka, Omega Tolessa, Markos Tesfaye, Kristina Adorjan, Wolfgang Krahl, Elias Tesfaye, Yimenu Yitayih, Ralf Strobl, Eva Grill.

**Visualization:** Matiwos Soboka.

**Writing – original draft:** Matiwos Soboka.

**Writing – review & editing:** Matiwos Soboka, Omega Tolessa, Markos Tesfaye, Kristina Adorjan, Wolfgang Krahl, Elias Tesfaye, Yimenu Yitayih, Ralf Strobl, Eva Grill.

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
