## [Decision Letter · Decision Letter 0]

11 Mar 2020

PONE-D-20-01650

Factors associated with khat use among patients with tuberculosis in Southwest Ethiopia: A longitudinal  study

PLOS ONE

Dear Dr. Soboka,

Thank you for submitting your manuscript to PLOS ONE. After careful consideration, we feel that it has merit but does not fully meet PLOS ONE’s publication criteria as it currently stands. Therefore, we invite you to submit a revised version of the manuscript that addresses the points raised during the review process.

In addition, please adhere to the following reporting guidelines:

- https://www.equator-network.org/reporting-guidelines/strobe/

- https://www.equator-network.org/reporting-guidelines/espacomp-medication-adherence-reporting-guideline-emerge/

We would appreciate receiving your revised manuscript by 21/04/2020. To enhance the reproducibility of your results, we recommend that if applicable you deposit your laboratory protocols in protocols.io, where a protocol can be assigned its own identifier (DOI) such that it can be cited independently in the future. For instructions see: http://journals.plos.org/plosone/s/submission-guidelines#loc-laboratory-protocols

We look forward to receiving your revised manuscript.

Kind regards,

Tim Mathes

Academic Editor

PLOS ONE

Journal Requirements:

"Our gratitude is extended to Jimma University for funding the project. Similarly, we are thankful for IPPG for funding the project partially. Also, our gratitude extended to Dr. Michael Odenwald for contributing money from his pocket to support the project."

Reviewers' comments:

Reviewer's Responses to Questions

**Comments to the Author**

1. Is the manuscript technically sound, and do the data support the conclusions?

Reviewer #1: No

Reviewer #2: Partly

Reviewer #3: Partly

2. Has the statistical analysis been performed appropriately and rigorously? 

Reviewer #1: Yes

Reviewer #2: No

Reviewer #3: Yes

3. Have the authors made all data underlying the findings in their manuscript fully available?

Reviewer #1: Yes

Reviewer #2: No

Reviewer #3: Yes

4. Is the manuscript presented in an intelligible fashion and written in standard English?

Reviewer #1: Yes

Reviewer #2: Yes

Reviewer #3: Yes

5. Review Comments to the Author

Reviewer #1: Factors associated with Khat use among patients with tuberculosis in Southwest

Ethiopia: A longitudinal study by Motiwas Soboka et. al

This paper tries to explore the factors associated with Khat use among tuberculosis patients. However the paper has serious flaws of research design and rationale. It lacks coherency in terms of objectives of research. What factors authors are trying to associate??

Following are the specific comments.

1. The stated objective of the research is “there is limited information on factors associated with Khat use among patients with TB and this study aimed at addressing this research gap.” But data and conclusion doesn’t support it

2. Study design: The study design doesn’t support the curation of factors associated with Khat use.

3. Sample size: the sample size is not adequate enough to draw such broad conclusion.

4. The title and the subsequent research methodology adopted do not match at all.

5. There are language issues but they are of minor nature.

Reviewer #2: Study title: Factors associated with khat use among patients with tuberculosis in Southwest Ethiopia: A longitudinal study

General comment:

The paper is generally well-written and adds to the broader literature on substance use and TB in Africa. The findings may help readers better understand common factors that contribute to Khat use in this sub group; these insights may contribute to interventions. Although the paper may address a gap, there are several major weaknesses and the paper could be improved with a stronger background section, further elaboration of the methods, and a stronger and more well-organized discussion of the results. Specific comments follow:

Title & Introduction

The title is overly promising and yet the study mainly focuses on TB outpatients on DOTs. The authors should clarify the title further.

The burden of TB is fairly well described but the introduction is scanty of known risk factors for TB and khat use separately. In the discussion section, it might be worth highlighting if these risk factors are ant different from those found in this study, and why? This should be addressed.

Methods

Why was use of other illicit drug used a s an exclusion criteria?

The sources of selection of participants are clear but the and methods of both participant selection and follow up remain vague.

The authors need to clearly define khat use and to indicate sources of data and details of methods of assessment (measurement) and the rationale for the choice assessment. Indicate elsewhere in the paper that you refer to self-reported khat use not simply Khat use.

Your paper has many sources of selection bias from how participants were selected to how assessments of outcomes and exposures were done. Describe any efforts to address potential sources of bias.

Explain how the sample size was arrived at and the rationale underlying the models that were built during the analysis.

It remains unclear to me what conceptual framework underpinned the analysis, which confounders were known from before and how these were controlled for and why they were included.

Results

Would appreciate a note of missing data and how this was handled. At a minimum, Indicate the number of participants with missing data for each variable of interest

You need to show how the associations changed overtime-from baseline to endline and give plausible reasons.

Discussion

‘To our knowledge, this study is the first longitudinal study conducted among patients with tuberculosis investigating predictors of khat use in Africa’. Vague statement consider rephrasing.

Authors should discuss limitations of the study further, taking into account sources of potential bias or imprecision. Discuss both direction and magnitude of any potential bias.

Conclusion: ‘The findings imply that male patients and patients with alcohol use disorders should be focused on while screening for khat use among TB patients.’ I would argue that the goals of screening for khat use need to be much broader than male gender and AUD. Consider re phrasing this seemingly misleading statement.

Reviewer #3: Thank you for giving the opportunity to review this manuscript titled “Factors associated with khat use among patients with tuberculosis in Southwest Ethiopia: A longitudinal study” [PONE-D-20-01650]

Dear authors, Good study surely based on the needs of your area! Please consider the followings

1- Overall, this is an interesting study exploring the use of plants (khat) for stimulation purpose among patients with tuberculosis. However, it is unclear whether the use among the local population is mainly recreational, or due to its medicinal properties. Does Khat chewing has any scientific explanation to causes addiction?

2- Dear authors, I am great full for your efforts. But, your main massage is to demonstrate khat use by focusing on epidemiological data and factors associated with it among tuberculosis patients. However, as it is explained in your background information, chewing khat is not uncommon including tuberculosis patients. If so, I believe that the big deal is not reporting the magnitude or level of khat use, rather, since your study is longitudinal, the better was to give important intervention and to see its effect. This might helpful to recommend for policy makers and other stakeholders at national level. –

otherwise, you should write more about the gap in your context and the real problem which motives you to do such a study. Problem statement is not convincing! The clear gap to see the prevalence of khat use and factors affecting it at different stages is not well explained.

3- say the evidences that emphasize on the importance of to take preventive measures and reduces medical complications related tuberculosis that might be caused by Khat consumption during medication follow-up apart from non-adherence (found in your study)?

4- In your study, there is nothing intervention or education is given for participants i.e. not indicated in your writing, however, varied findings are documented regarding the prevalence of khat use at baseline and last data collection time. What sort of things can be reasons to decrease the prevalence (in percentile)? And writ more about this

5- This study reported that the use of Khat among patients with tuberculosis & alcohol problem is higher. Please discuss further whether this something good or bad, because it may divert the individuals from using other more harmful substances.

6- In your exclusion criteria………polysubstance users (using two or more substances) and patients were excluded from the study. However, alcohol use problem (disorder) is a predictor to use khat. It is unclear and ambiguous idea. Could you say more about this please? (Since the participants in your study were using both khat and alcohol)

7- Also in your sampling procedure regarding to exclusion and inclusion criteria, it is stated as…..We included all patients who had recently been diagnosed with TB who had started DOT in the selected health centers and hospitals. Patients were included only if they had started TB treatment within less than four weeks before inclusion and were not planning to transfer to other health institutions. Patients aged less than 18 years, patients infected with multidrug-resistant TB strains ,………..were excluded from the study. Reason behind to include recently diagnosed (4 weeks) only?? Does it have any significance difference to chew khat with people living with TB for long?????

8- In your data collection procedure part, It is first time Agaro is mentioned. No context provided as to why pre-testing was done in Agaro (which reader outside Ethiopia may not know why it is important for a study). Write more about Agaro where it is located??

9- Doe the pretest was done using both version of the questionnaires (Afan oromo and Amharic). ???

10- Self-Reporting Questionnaire (SRQ-20) to assess mental distress, -WHO's structured questionnaire of Alcohol Use Disorder Identification Test (AUDIT) to asses Alcohol Use Disorders and questions used to assess the frequency and patterns of Khat use- Operational definitions should be about measures items, dimensions, rating scales, theoretical basis, validity, reliability and so on...

I thought it seems to write it again deeply to increase the quality.

11- Prevalence of khat use at baseline and last assessment examined different (39.2% Vs 37.3%). What it its implication?? Clearly not explained specifically among tuberculosis patients? I suggest to discuss it more including thee pattern of khat chewing.

12- To show factors associated with khat chewing I suggest to include confidence interval also so that it will be clear more.

13- Try to discuss your findings detail……….. About merchants (why they chew more….. the prevalence of khat use if there is study findings done longitudinally…..)??? discussion is limited.

14- Lastly …I thought it looks scientifically sound. If your title re written as “Magnitude and factors associated with khat use among patients with tuberculosis in Southwest Ethiopia: A longitudinal study”

6. PLOS authors have the option to publish the peer review history of their article (what does this mean?). If published, this will include your full peer review and any attached files.

Reviewer #1: No

Reviewer #2: No

Reviewer #3: No

---

## [Author Response · Author response to Decision Letter 0]

23 Apr 2020

Response to reviewers

Reviewer #1

This paper tries to explore the factors associated with Khat use among tuberculosis patients. However the paper has serious flaws of research design and rationale. It lacks coherency in terms of objectives of research. What factors authors are trying to associate??

Following are the specific comments.

1. The stated objective of the research is “there is limited information on factors associated with Khat use among patients with TB and this study aimed at addressing this research gap.” But data and conclusion doesn’t support it

Response: Indeed we wanted to examine predictors of Khat use longitudinally, and we apologize if this was not completely clear. Our results showed that being male, being a government employee and the presence of alcohol use disorders were associated with khat use among TB patients. To our knowledge, this is one of the few studies that investigated a broad range of potential predictors. 

2. Study design: The study design doesn’t support the curation of factors associated with Khat use.

Response: Data originates from a comprehensive cohort study, so the longitudinal aspect was incorporated. Cohort studies are generally state-of-the-art for investigating risk factors.

3. Sample size: the sample size is not adequate enough to draw such broad conclusion.

Response: This is a valuable comment. We now mention this as a limitation of the study. The sample size was calculated based on previous studies with a prevalence of almost 50%. We found a prevalence of about 40%, a difference which is acceptable. For regression models, 10 cases (here: patients using khat) per predictor to be included is generally sufficient. Thus, our sample size allows models with up to 10 predictors which is well in line with our specifications. 

4. The title and the subsequent research methodology adopted do not match at all.

Response: We are grateful for this comment and amended the title. 

5. There are language issues but they are of minor nature.

Response: We edited the manuscript. 

Reviewer #2

General comment:

The paper is generally well-written and adds to the broader literature on substance use and TB in Africa. The findings may help readers better understand common factors that contribute to Khat use in this sub group; these insights may contribute to interventions. Although the paper may address a gap, there are several major weaknesses and the paper could be improved with a stronger background section, further elaboration of the methods, and a stronger and more well-organized discussion of the results. 

Specific comments follow:

Title & Introduction

1. The title is overly promising and yet the study mainly focuses on TB outpatients on DOTs. The authors should clarify the title further. 

Response: We agree and have amended the title. However, the reason why we have included only DOTs patients is that there is no chance that patients who are getting treatment at the inpatient department would access substance including khat. Also, we have included this issue in limitation of the study stating that the findings of this study will not be generalized to those patients who are getting treatment at the inpatients department. 

2. The burden of TB is fairly well described but the introduction is scanty of known risk factors for TB and khat use separately. In the discussion section, it might be worth highlighting if these risk factors are ant different from those found in this study, and why? This should be addressed.

Response: We have accepted the reviewer’s comment and amended the introduction and discussion. 

##Methods

3. Why was use of other illicit drug used a s an exclusion criteria?

 Response: We did not exclude the use of illicit drugs but unfortunately we could not find a patient who uses illicit drug and this could be related to availability or fear of legal consequences. So, we consider this as social desirability and include it in the limitation of the study. 

4. The sources of selection of participants are clear but the and methods of both participant selection and follow up remain vague.

Response: We have accepted the comment and amend the manuscript. 

4. The authors need to clearly define khat use and to indicate sources of data and details of methods of assessment (measurement) and the rationale for the choice assessment. Indicate elsewhere in the paper that you refer to self-reported khat use not simply Khat use.

Response: We have accepted the comment and amend the manuscript.

5. Your paper has many sources of selection bias from how participants were selected to how assessments of outcomes and exposures were done. Describe any efforts to address potential sources of bias. 

Response: We are grateful for the reviewer's comment. Since we have selected the participants consecutively participants who have fulfilled the inclusion criteria had an equal chance to be involved in the study. So, there was a limited selection and assessment of outcomes and exposures bias. 

6.Explain how the sample size was arrived at and the rationale underlying the models that were built during the analysis.

Response: We have accepted the comment and amended the manuscript. 

7.It remains unclear to me what conceptual framework underpinned the analysis, which confounders were known from before and how these were controlled for and why they were included.

Response: The variables with potential confounders were controlled using multivariable modeling in which we have included potential variables that could affect khat use. We used risk factors that were known from the literature. Therefore, the findings have been adjusted for potential confounders. For example, gender and age and others. 

 ##Results

8.Would appreciate a note of missing data and how this was handled. At a minimum, indicate the number of participants with missing data for each variable of interest

Response: We have accepted the comment and amended the manuscript. We have used the case-wise deletion of participants with missing data. Hence, 22 participants have been omitted from the analysis because of missing data on one or more variables.

9. You need to show how the associations changed overtime-from baseline to endline and give plausible reasons.

Response: We have amended the manuscript based on the reviewer's comment. As it is indicated in table 2 the strength of the association was improved over time as witnessed by further improvement of the model fit (BIC=815.7). 

##Discussion

10.‘To our knowledge, this study is the first longitudinal study conducted among patients with tuberculosis investigating predictors of khat use in Africa’. Vague statement consider rephrasing.

Response: We have accepted the comment. 

Authors should discuss limitations of the study further, taking into account sources of potential bias or imprecision. Discuss both direction and magnitude of any potential bias.

Response: We are grateful for the reviewer's comment and added some additional potential limitations. Also, we want to inform the reviewer as we have reported the following bias: social desirability bias, data collection by health professionals working in the clinic, tools used to assess reported khat use etc. 

Conclusion: ‘The findings imply that male patients and patients with alcohol use disorders should be focused on while screening for khat use among TB patients.’ I would argue that the goals of screening for khat use need to be much broader than male gender and AUD. Consider re phrasing this seemingly misleading statement.

Response: We agree with the reviewer's comment that screening of patients for khat use has to be comprehensive, but our conclusion is only based on the findings of the study. We rephrased the sentence with caution. The conclusion now reads as ”The findings indicate screening for khat use among TB patients needs to prioritize males and those with alcohol use disorders.”

Reviewer #3

Thank you for giving the opportunity to review this manuscript titled “Factors associated with khat use among patients with tuberculosis in Southwest Ethiopia: A longitudinal study” [PONE-D-20-01650]

Dear authors, Good study surely based on the needs of your area! Please consider the followings

1. Overall, this is an interesting study exploring the use of plants (khat) for stimulation purpose among patients with tuberculosis. However, it is unclear whether the use among the local population is mainly recreational, or due to its medicinal properties. Does Khat chewing has any scientific explanation to causes addiction?

Response: Thank you for the constructive comment. Khat can cause psychological dependence among long time users and we have included this in the introduction. Also, local people use khat for recreational, social lubrication, to reduce the feeling of hunger, as self-medication and to stay alert while praying. We have included some of these factors while we were discussing our results. 

2. Dear authors, I am great full for your efforts. But, your main massage is to demonstrate khat use by focusing on epidemiological data and factors associated with it among tuberculosis patients. However, as it is explained in your background information, chewing khat is not uncommon including tuberculosis patients. If so, I believe that the big deal is not reporting the magnitude or level of khat use, rather, since your study is longitudinal, the better was to give important intervention and to see its effect. This might helpful to recommend for policy makers and other stakeholders at national level. –otherwise, you should write more about the gap in your context and the real problem which motives you to do such a study. Problem statement is not convincing! The clear gap to see the prevalence of khat use and factors affecting it at different stages is not well explained.

Response: We have accepted about explaining the statement of the problem further and amended the manuscript. But, arguably, there are not many studies targeting khat use among patients with tuberculosis. There is also a scarcity of information regarding the magnitude of khat use from longitudinal studies as many of the studies are cross-sectional. Also, those studies did not assess khat use primarily considering it as the main outcome variable but they report khat use on their way to assess other problems among TB patients. Our data provide evidence for continued use of khat among TB outpatients despite regular contact with health professionals so that we make the case for more structured and effective interventions to be developed and tested. Regrettably, our study was not designed as an intervention study, but the reviewer is correct to state that this should be the aim of further research, and this study is most likely to inform future trials in this regard. 

3. Say the evidences that emphasize on the importance of to take preventive measures and reduces medical complications related tuberculosis that might be caused by Khat consumption during medication follow-up apart from non-adherence (found in your study)?

Response: We have studied the effect of khat use among tuberculosis including non-adherence but we have prepared a separate manuscript which gives detail of this information. 

4. In your study, there is nothing intervention or education is given for participants i.e. not indicated in your writing, however, varied findings are documented regarding the prevalence of khat use at baseline and last data collection time. What sort of things can be reasons to decrease the prevalence (in percentile)? And writ more about this

Response: We have accepted the comment partly and amend the manuscript accordingly. Our study is not an interventional study so we did not primarily focus on intervention, but we have trained our data collectors to give education regarding problematic khat use and its effect at the end of data collection. During data collection, we have advised those who use khat more frequently to talk to a mental health professional and we have included this under the topic of “ethical consideration”. We added the probable reason why khat use was declined over time to the discussion part. 

5. This study reported that the use of Khat among patients with tuberculosis & alcohol problem is higher. Please discuss further whether this something good or bad, because it may divert the individuals from using other more harmful substances.

Response: We have accepted the comment and made discussion as follows: Combining alcohol and khat would affect patients' treatment outcomes, and lead to physical and mental health problems. Because both khat and alcohol were found to have a potential impact on the immunity of the user and associated with mental distress so that patients may develop severe medical and mental health complications or die earlier than non-users. Furthermore, these two substances have an association with treatment-resistant tuberculosis

6. In your exclusion criteria………polysubstance users (using two or more substances) and patients were excluded from the study. However, alcohol use problem (disorder) is a predictor to use khat. It is unclear and ambiguous idea. Could you say more about this please? (Since the participants in your study were using both khat and alcohol)

Response: We are grateful for the comment. We have excluded patients who were using three or more substances (polysubstance users) and amended the manuscript accordingly. We are sorry for the typing error.

7. Also in your sampling procedure regarding to exclusion and inclusion criteria, it is stated as…..We included all patients who had recently been diagnosed with TB who had started DOT in the selected health centers and hospitals. Patients were included only if they had started TB treatment within less than four weeks before inclusion and were not planning to transfer to other health institutions. Patients aged less than 18 years, patients infected with multidrug-resistant TB strains ,………..were excluded from the study. Reason behind to include recently diagnosed (4 weeks) only?? Does it have any significance difference to chew khat with people living with TB for long?????

Response: Since we were also following other outcomes such as adherence, mental health and quality of life, we have decided to include only new patients because it allows us to see any change over time. Also, the phase of treatment is important because in the first few weeks since patients have serious symptoms of tuberculosis we want to see whether patients increase or decrease their khat consumption. Furthermore, it is difficult to include other patients such as those who are on retreatment and MDR-TB because the duration of treatment is quite different and we have included this in the limitation of the study. 

8. In your data collection procedure part, It is first time Agaro is mentioned. No context provided as to why pre-testing was done in Agaro (which reader outside Ethiopia may not know why it is important for a study). Write more about Agaro where it is located??

Response: We have accepted the comment. 

9. Doe the pretest was done using both version of the questionnaires (Afan oromo and Amharic). ???

Response: Yes we have done the pretest and data collection in both languages because our participants speak either of the two languages. 

10. Self-Reporting Questionnaire (SRQ-20) to assess mental distress, -WHO's structured questionnaire of Alcohol Use Disorder Identification Test (AUDIT) to asses Alcohol Use Disorders and questions used to assess the frequency and patterns of Khat use- Operational definitions should be about measures items, dimensions, rating scales, theoretical basis, validity, reliability and so on...

I thought it seems to write it again deeply to increase the quality.

Response: We have accepted the comment and added detail information about the tools into the manuscript. 

11. Prevalence of khat use at baseline and last assessment examined different (39.2% Vs 37.3%). What it its implication?? Clearly not explained specifically among tuberculosis patients? I suggest to discuss it more including thee pattern of khat chewing.

Response: We have accepted the comment and amended the manuscript. 

12. To show factors associated with khat chewing I suggest to include confidence interval also so that it will be clear more.

Response: We have accepted the comment and included confidence intervals.

13. Try to discuss your findings detail……….. About merchants (why they chew more….. the prevalence of khat use if there is study findings done longitudinally…..)??? discussion is limited. 

Response: We have accepted the comment and added the explanation why merchant chew khat more. 

14. Lastly …I thought it looks scientifically sound. If your title re written as “Magnitude and factors associated with khat use among patients with tuberculosis in Southwest Ethiopia: A longitudinal study”

Response: We have accepted the comment and amended the title.

---

## [Decision Letter · Decision Letter 1]

26 May 2020

PONE-D-20-01650R1

Magnitude and predictors of khat use among patients with tuberculosis in Southwest Ethiopia: A longitudinal  study

PLOS ONE

Dear Dr. Soboka,

Thank you for submitting your manuscript to PLOS ONE. After careful consideration, we feel that it has merit but does not fully meet PLOS ONE’s publication criteria as it currently stands. Therefore, we invite you to submit a revised version of the manuscript that addresses the points raised during the review process.

Before we can accept the paper, English language editing is necessary. 

We look forward to receiving your revised manuscript.

Kind regards,

Tim Mathes

Academic Editor

PLOS ONE

Reviewers' comments:

Reviewer's Responses to Questions

**Comments to the Author**

1. If the authors have adequately addressed your comments raised in a previous round of review and you feel that this manuscript is now acceptable for publication, you may indicate that here to bypass the “Comments to the Author” section, enter your conflict of interest statement in the “Confidential to Editor” section, and submit your "Accept" recommendation.

Reviewer #1: All comments have been addressed

Reviewer #2: (No Response)

2. Is the manuscript technically sound, and do the data support the conclusions?

Reviewer #1: (No Response)

Reviewer #2: Partly

3. Has the statistical analysis been performed appropriately and rigorously? 

Reviewer #1: Yes

Reviewer #2: Yes

4. Have the authors made all data underlying the findings in their manuscript fully available?

Reviewer #1: Yes

Reviewer #2: No

5. Is the manuscript presented in an intelligible fashion and written in standard English?

Reviewer #1: Yes

Reviewer #2: No

6. Review Comments to the Author

Reviewer #1: In view of the comments raised by the reviewer and it's subsequent clarification and modification in the manuscript draft the reviewer suggest that the manuscript may be accepted in its revised form . However reviewer still believes that authors may further look for broader range of predictors ranging from social norms , anxiety and economical distress , changing cultural practices , and off course the medical model offering TB care , also if the whole of Ethiopia can be targetted with modifications in design and methodology so that broader meaning full conclusions can be drawn which ultimately can be useful for the policy makers , stakeholders and government institutions to better understand predictors for khat use among TB patients in whole of Ethiopia.

Reviewer #2: Dear author,

Thanks for sharing the revised paper and rebuttal.

I have further comments on the response attached.

Of concern is that there are major comments that remian unadrressed. In addition, the paper would use a review from a native English speaker.

As is, this paper is not suitable for publication untill these issuse are clearly addresed. I now leave the final decision on this paper to the editor.

7. PLOS authors have the option to publish the peer review history of their article (what does this mean?). If published, this will include your full peer review and any attached files.

Reviewer #1: No

Reviewer #2: No

---

## [Author Response · Author response to Decision Letter 1]

27 Jun 2020

Language: This manuscript is edited by a professional English language editor. 

Response to reviewer

Comment [A1]: Could shorten title further:

“Magnitude and predictors of khat use among tuberculosis patients in Southwest Ethiopia: A longitudinal study”

Response: Thank you for the comment. We would like to keep the current title because we are concerned that further shortening of the title may make it less informative about the objectives, the area and design of the study.

Comment [A2]: Limited chance perhaps?

Response: We agree with the reviewer's comment because there is limited chance for TB inpatients access to substances including khat. We have included this issue in the discussion section on page 18, line 12 to 14 .

Comment [A3]: This change doesn’t show in the limitations section. Please revise.

Response: Thank you for your comment. This information was stated on page 18, lines 4 to 7 on the previous version of the manuscript. Now we have separately reported it on page 18, line 9 to 10 as “The findings of this study can not be generalized to those patients who are getting treatment at the inpatient facilities”

Comment [A4]: This response is unsatisfactory.

Response: On the top of theoretical importance and the adequate number of participants in each cell for each category, we have also used literature, clinical knowledge, and experience to build the model. The reason why we have included confounders is that they have the potential to distort the association between dependent and independent variables. Also, it is difficult to identify variables that independently predict khat use without controlling for confounders. Therefore, we have controlled them using multivariable analysis. We have made additional clarification to the analysis on page 9 lines 3 to 4.

Comment [A5]: Please show this missing data as a footnote to the table

Response: we have accepted and made changes on page 11.

Comment [A6]: Consider: “To our knowledge, this study is the first longitudinal study investigating predictors of khat use conducted among patients with tuberculosis in Africa.”

Response: We have accepted the comment and made changes on page 15, lines 1 and 2.

Comment [A7]: The author does not address issues of the direction of potential biases listed. This is a major omission in my view.

Response: We have reported the direction of bias on 18, lines 2 to 4 and now read as “Due to social desirability, patients may minimize or deny their khat consumption, which could have a negative impact on the extent of khat use. It could also affect the association of predictors with the outcome variables. In addition, health professionals working in a TB clinic might have contribute to this bias, as patients may be inclined not to report their khat use or to minimize its amount.

Comment [A8]: From your findings, it seems that screening for Khat is a major conclusion itself. Paying attention wo men and people with AUD is secondary right?

Response: We have rephrased the conclusion as “All patients diagnosed with TB should be screened for khat use and a particular emphasis should be given to males and individuals with a history of alcohol use” The changes are available on page 2, lines 2 and 3 and on page 18, lines 4 and 5.”

---

## [Editor Report · Decision Letter 2]

1 Jul 2020

Magnitude and predictors of khat use among patients with tuberculosis in Southwest Ethiopia: A longitudinal  study

PONE-D-20-01650R2

Dear Dr. Soboka,

We’re pleased to inform you that your manuscript has been judged scientifically suitable for publication and will be formally accepted for publication once it meets all outstanding technical requirements.

Kind regards,

Tim Mathes

Academic Editor

PLOS ONE
---

## [Editor Report · Acceptance letter]

20 Jul 2020

PONE-D-20-01650R2 

Magnitude and predictors of khat use among patients with tuberculosis in Southwest Ethiopia: A longitudinal study 

Dear Dr. Soboka:

I'm pleased to inform you that your manuscript has been deemed suitable for publication in PLOS ONE. Congratulations! Your manuscript is now with our production department. 

Kind regards, 

on behalf of

Dr. Tim Mathes 

Academic Editor

PLOS ONE